

# Rural micro-credit model design and credit risk assessment via improved LSTM algorithm

Xia Gao, Xiaoqian Yang and Yuchen Zhao

Business School, University of Jinan, Jinan, Shandong, China

## ABSTRACT

Rural microcredit plays an important role in promoting rural economic development and increasing farmers' income. However, traditional credit risk assessment models may have insufficient adaptability in rural areas. This study is based on the improved Long Short Term Memory (LSTM) algorithm using self organizing method, aiming to design an optimized evaluation model for rural microcredit risk. The improved LSTM algorithm can better capture the long-term dependence between the borrower's historical behavior and risk factors with its advantages in sequential data modeling. The experimental results show that the rural microcredit risk assessment model based on the self organizing LSTM algorithm has higher accuracy and stability compared to traditional models, and can effectively control credit default risk, providing more comprehensive risk management support for financial institutions. In addition, the model also has real-time monitoring and warning functions, which helps financial institutions adjust their decisions in a timely manner and reduce credit losses. The practical application of this study is expected to promote the stable development of rural economy and the advancement of financial technology. However, future work needs to further validate the practical application effectiveness and interpretability of the model, taking into account the special circumstances of different rural areas, in order to achieve sustainable application of the model in the rural microcredit market.

## INTRODUCTION

Rural microcredit is an important component of the rural financial system, which can effectively alleviate the funding needs in the agricultural production process and increase their income (*Wang & Li, 2023*; *Ma et al., 2023*). However, due to the current information asymmetry problem in the credit loan market, most commercial banks have certain deficiencies in the risk management of rural micro credit loans (*Murta & Gama, 2022*; *Aramonte, Lee & Stebunovs, 2022*). In terms of the credit risk management of farmers' microfinance, the low accuracy of farmers' credit risk prediction and the lack of scientific credit risk assessment system have all brought significant credit risk risks to farmers' microfinance business (*Alhassan et al., 2023*; *Lu et al., 2022*; *Alemu & Zerhun, 2023*). In addition, due to the fact that the economic activities in rural areas are mostly agricultural

Corresponding author
Xiaoqian Yang,
yangxiaoqian112233@163.com

production, there are certain natural and market risks. Small loans often face high default risks, so how to effectively manage loan risks is a challenge. To address the lack of credit and risk assessment in the microlending process for farmers, which is crucial for the regulation of financial services, it is urgently necessary to build a trustworthy credit and risk assessment system.

The majority of credit risk assessment models have caught the interest of many academics in recent years, and these models are built utilizing statistical and machine learning techniques. When the feature space is big, the accuracy of conventional random forest and logistic regression approaches is poor (*Zheng, Liu & Ge, 2022*; *Leonard et al., 2022*; *Oh et al., 2022*). Convolutional neural networks is also commonly used for speech recognition software and computer vision simulation (*Zhao, 2022*; *Dua et al., 2022*). The rapid growth of big data has accelerated the advancement of neural networks. There has been tremendous advancement in a variety of classification and regression research results in sectors relevant to convolutional neural networks.

Methods including artificial neural networks, genomic planning, support vector machines, logical regression, and some hybrid models have achieved significant advancements in the field of credit risk assessment in terms of performance and accuracy. Many great algorithms and research techniques have been employed in the field of credit risk assessment for many trials over the past few years based on customer information data (*Machado & Karray, 2022*; *Shi et al., 2022*).

The number of neurons in the buried layer of neural networks is a challenging topic that never goes away. If there aren't enough neurons, the network can't be trained well and learning becomes challenging. Conversely, if there are too many neurons, overfitting is a common occurrence, which causes a significant increase in training time and complexity across the board. The prediction accuracy can be kept at a high level if the Long Short Term Memory (LSTM) network's hidden layer neuron count stays within a particular range (*Wu et al., 2021*; *Xu et al., 2022*). In order to obtain better prediction results, it is crucial to make sure the number of hidden layer neurons is within this range while building LSTM neural network models. This research employs a sensitivity-based self-organizing algorithm to automatically modify the number of hidden layer neurons during training in order to address the issue. Hidden layer neurons are created when the LSTM network's output falls short of the intended level, and they are removed when their sensitivity falls short of a predetermined threshold. To our best knowledge, the main contributions of this article are as follows

(i) Improved LSTM algorithm: The traditional LSTM performs well in sequence data modeling, but there may be some deficiencies in rural microfinance data modeling. The improved LSTM algorithm can propose new network structures or optimization strategies based on the characteristics of rural credit risk, in order to improve the performance and accuracy of the model.

(ii) Introducing rural credit risk factors: Rural credit risk assessment needs to consider many factors specific to rural areas, such as crop yields, weather effects, market fluctuations, *etc*. The improved model can introduce these rural credit risk factors and incorporate

them into the decision-making process of credit risk assessment, thereby improving the predictive ability of the model.

(iii) Real time and dynamic updates: Rural credit risk assessment needs to have the characteristics of real-time and dynamic updates. The improved model can design corresponding mechanisms to achieve real-time updates of the model to adapt to changes in rural credit risks.

As a result, this chapter provides a thorough introduction to the sensitivity algorithm and suggests an LSTM model based on a self-organizing algorithm. Finally, experiments are used to validate this model's prediction performance.

## RELATED WORK

Credit risk assessment is the earliest developed financial risk management tool, and it is also the most successful link in the application of statistics and operational research in the financial industry. Generally, scoring techniques are used to comprehensively evaluate the credit performance of borrowers in terms of future repayment, *etc*. By using disciplines such as statistics, operational research, and data mining technology, the basic attribute characteristics, historical credit records based on systematic analysis of massive data such as behavioral information. By mining the credit and behavioral characteristics present in these data, creating a prediction model, quantitatively calculating the evaluation object's default probability, or assigning specific credit evaluation scores, one can determine the relationship between future credit performance and historical past records and determine the evaluation object's risk profile.

Model construction and data analysis are, in general, the two basic techniques used in data research (*Xu, Yan & Cui, 2022*; *Zhang et al., 2022*). According to the author's empirical findings in *Collier & Hampshire (2010)*, the higher the ratio of the loan amount to annual income, in terms of credit evaluation indicators, the lower the possibility of securing a loan and the higher the loan interest rate. In *Lin, Li & Zheng (2017)*, empirical research techniques list a number of particular evaluation indicators, such as gender, age, marital status, loan size, and so forth.

Banks can evaluate the risks that loan applicants pose using credit scoring. In other terms, it enables financial institutions to vouch for the legitimacy of loan seekers (*Ping, 2016*). Credit institutions always work to reduce the chance that borrowers won't be able to repay their loans. Therefore, based on specific criteria, credit scoring enables banks to mitigate these risks. The reliability and financial stability of borrowers are relevant to the stated criteria. Relevant academics have conducted a number of studies to evaluate credit risk. In *Malekipirbazari & Aksakalli (2015)*, the author developed a credit risk assessment model using decision support tools, with the major determinants being loan interest rates, the annual income of the lender, and the length of the borrower's repayment period. In *Rajamohamed & Manokaran (2018)*, the author examined how social ties affected credit appraisal and discovered that both borrower borrowing success and default risk are significantly influenced by social ties. Loan groups and more soft information do help to lower the credit risk associated with online lending, but as loan group leaders frequently

have a monopoly rent due to their access to more information and decision-making authority, this raises borrower costs and default risk. Additionally, soft information may also reveal the educational background of the lenders.

A new combination strategy based on classifier consistency to combine several classifiers was proposed in *Ala'raj & Abbod (2016)* for credit evaluation. Through the use of credit score data, the author confirmed the model's good predictive performance. One of the algorithms used to determine credit risk is the random forest algorithm. According to research, the predictions made using random forests are highly precise and accurate. Relevant academics suggested a continuous rule extraction strategy for credit scoring in order to increase the model's precision and interpretability.

According to pertinent research, it is currently impossible to accurately and completely explain the factors that influence borrowers' default, and a lack of understanding of the data has caused the model's prediction accuracy to be low. As a result, it is challenging to meet the demands of big data for credit analysis and identification of borrowers. In order to create credit risk assessment models fast and reliably, the improved LSTM method proposed in this article.

Firstly, we will provide a detailed description of the architecture of the proposed self-organizing LSTM risk assessment model. This includes the composition of the input layer, hidden layer, and output layer of the model, as well as the specific parameter settings for each layer. We will explain how self-organizing algorithms automatically adjust the number of neurons and why they introduce rural specific risk factors and multi-level feature learning mechanisms. At the same time, we will describe the implementation method of real-time monitoring and early warning, as well as how the model dynamically updates the borrower's risk assessment.

Then, we will provide a detailed introduction to the experimental setup, including the selection and preprocessing of the dataset, the partitioning of the training set, validation set, and test set, as well as the selection of other neural network algorithms for comparison. We will describe the training process and hyperparameter adjustment methods of the model, and explain the selection and significance of indicators for model evaluation.

Finally, we will list the performance of the model on evaluation indicators such as accuracy, precision, recall, F1 score, and AUC value, and compare it with other neural network algorithms. We will focus on analyzing the advantages of self-organizing LSTM algorithm in these indicators, as well as how to solve the problems of inaccurate credit ratings and insufficient risk estimation. We will also discuss the performance of the model at different risk levels and explain how the features and mechanisms in the model can help improve accuracy and estimation.

## LSTM ALGORITHM BASED ON SELF-ORGANIZATION

### LSTM algorithm

To address RNN's gradient disappearance issue, *Hochreiter & Schmidhuber (1997)* introduced LSTM as a specific circulating neural network. Three control gates were first introduced by the network, which also established a node structure distinct from that

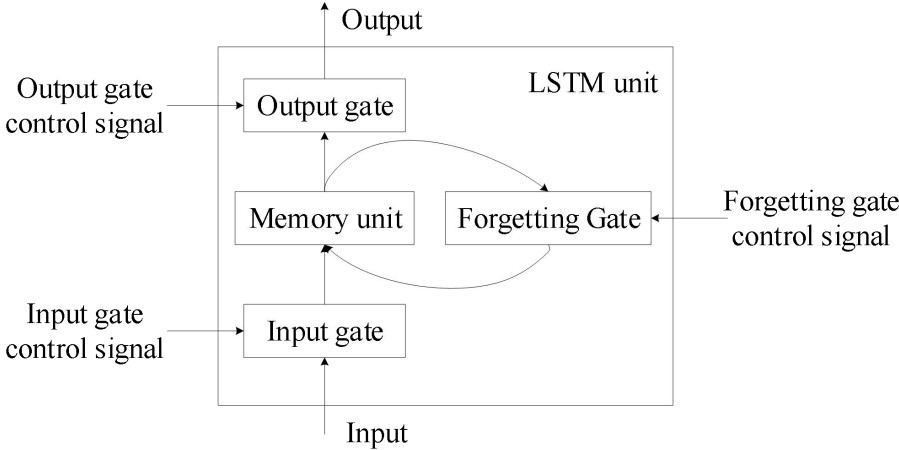

**Figure 1  Overall structure of the LSTM unit.**

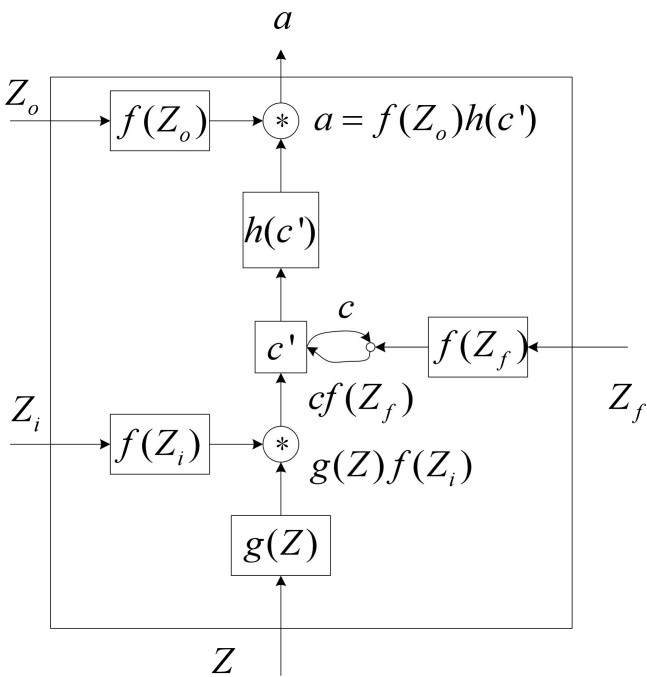

**Figure 2  Internal structure diagram of LSTM unit.**

of regular neurons. Figure 1 depicts the general structure of this node, which is referred to as an LSTM unit.

The four input components of the LSTM unit are input, control signals for input gates, forgetting gates, and output gate control signals. LSTM cells have a parameter amount that is four times greater than that of regular neurons. Figure 2 depicts a more thorough internal structure so that the LSTM unit's operating principle can be understood in greater depth.
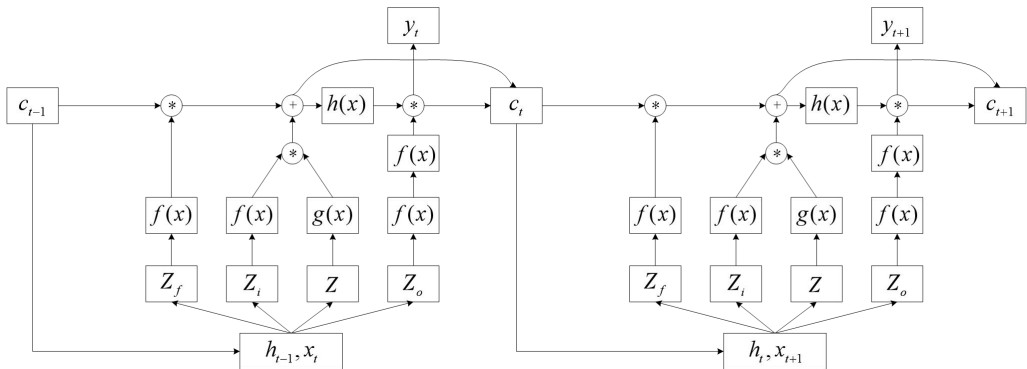

**Figure 3   Overall structure diagram of LSTM network.**

Where $Z$ is input, $Z_i$ is the input gate control signal, $Z_f$ is the forgetting gate control signal, and the $Z_o$ is the output gate control signal. The $f(x)$ function is typically a sigmoid function

$$f(x) = \frac{1}{1+e^{-x}}. \tag{1}$$

The value between [0,1] that represents the door's opening degree can be controlled by the sigmoid function. The activation functions g(x) and h(x) are identical.

First, $Z$ gets $g(Z)$ through the activation function g(x), $Z_i$ gets $g(Z_i)$ through the sigmoid function, and $Z_f$ gets $g(Z_f)$ through the sigmoid function. $c'$ gets $h(c')$ through the activation function $h(x)$, $Z_o$ gets $g(Z_o)$ through the sigmoid function.

Compared to RNN, LSTM networks replace hidden layer neurons in RNN with LSTM cells, resulting in a fourfold increase in input parameters and a change in output format. Put the LSTM unit into the entire network structure, and the computing structure is shown in Fig. 3.

Where $x_t$ and $h_{t-1}$ together serve as input portions of the hidden layer at time $t$, multiply different weight vectors and pass through the activation function to obtain the control signals $Z_f$, $Z_i$, and $Z_o$ of the three gates, and the input value $Z$. As a result

$$
\begin{aligned}
Z_f &= W_f \cdot [h_{t-1}, x_t] + b_f \\
Z_i &= W_i \cdot [h_{t-1}, x_t] + b_i \\
Z_o &= W_o \cdot [h_{t-1}, x_t] + b_o \\
Z &= W \cdot [h_{t-1}, x_t] + b_x
\end{aligned} \tag{2}
$$

where $b_f$, $b_i$, $b_o$, and $b_x$ are offsets for different connection weights. After a series of operations by the LSTM unit, the value c stored in the memory unit is updated, have

$$c' = g(W_x \cdot [h_{t-1}, x_t] + b_x) f(W_i \cdot [h_{t-1}, x_t] + b_i) + cf(W_f \cdot [h_{t-1}, x_t] + b_f). \tag{3}$$

As seen above, in LSTM neural networks, the historical information that affects the subsequent instant includes both the values stored in the memory cells of the LSTM cells as

well as the output of the hidden layer neurons at the preceding moment. When compared to RNN, LSTM neural networks can remember more historical knowledge, minimize gradient disappearance, and better suit the trend of long-term time series data.

## Sensitivity calculation

This article uses an algorithm based on sensitivity to conduct self organizing training for LSTM neural networks. The red arrow indicates the self feedback part of the hidden layer neuron, that is, the output of the hidden layer neuron at time $t$, *i.e.,* $z_1(t) = (H_1(t-1), H_2(t-2), \ldots, H_N(t-1))$, and the blue arrow indicates the output portion from the hidden layer to the output layer, that is, the output of the hidden layer neuron at time $t$, *i.e.,* $z_2(t) = (H_1(t), H_2(t), \ldots, H_N(t))$.

The concept of sensitivity is as follows:

$$S_h = \frac{Var[E(Y|Z_h)]}{Var(Y)}, (h=1,2,\ldots,N) \tag{4}$$

where $Z_h$ represents the h-th input factor, $Y$ is the output layer output of the model, A represents the expected output of $Y$ at a fixed value of $hZ$, and $X$ represents the calculated variance.

For the LSTM neural network model constructed in this article, sensitivity analysis is divided into two calculation parts: indirect sensitivity and direct sensitivity. The red self feedback section in Fig. 4.1 is used to calculate indirect sensitivity, while the blue output section is used to calculate direct sensitivity. Therefore, according to Formula (4), the calculation formula for indirect sensitivity is as follows:

$$S_h^1(t) = \frac{Var_h[E(Y(t)|Z_h^1 = H_h(t-1))]}{Var(y(t))} \tag{5}$$

where $H_h(t-1)$ represents the output of the h-th hidden layer neuron at time $t-1$ and $y(t)$ represents the output of the network output layer. We have

$$
\begin{aligned}
H_h(t-1) &= h_{t-1}(c')f_{t-1}(Z_o) \\
&= h(g(W_x \cdot [H_h(t-2), x_t] + b_x)f(W_i \cdot [H_h(t-2), x_t] + b_i) \\
&\quad + cf(W_f \cdot [H_h(t-2), x_t] + b_f)) \times f(W_o \cdot [H_h(t-2), x_t] + b_o)
\end{aligned} \tag{6}
$$

where $W_x, W_i, W_f$ and $W_o$ represent the unit input weight, the weight of the input gate control signal, the weight of the forgotten gate control signal, and the weight of the output gate control signal, respectively. $b_x, b_i, b_f$ and $b_o$ represent input bias, input gate control signal bias, forgetting gate control signal bias, and output gate control signal bias, respectively.

The following is how the output of the output layer is represented.

$$y(t) = \sum_{j=1}^{N} W_j(t)H_j(t) \tag{7}$$

where $W_j$ represents the connection weight of the $j$th hidden layer neuron to the output layer at time $t$.
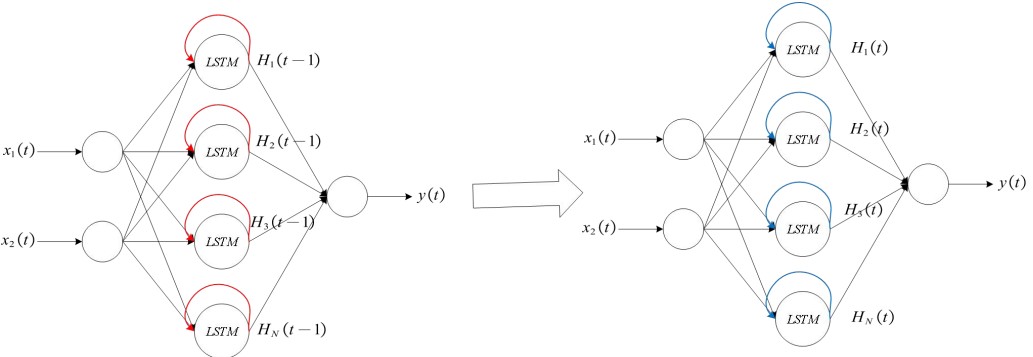

**Figure 4   Exploded View of LSTM Network.**

The direct sensitivity is calculated as follows:

$$S_h^2(t) = \frac{Var_h[E(y(t)|Z_h^2 = H_h(t))]}{Var(y(t))}. \tag{8}$$

The difference between direct sensitivity and indirect sensitivity is that the condition for obtaining the desired output is replaced by the output of the h-th hidden layer neuron at time $t$. Then, we have

$$
\begin{aligned}
H_h(t) &= h_t(c')f_t(Z_o) \\
&= h(g(W_x \cdot [H_h(t-1), x_t] + b_x) f(W_i \cdot [H_h(t-1), x_t] + b_i) \\
&\quad + cf(W_f \cdot [H_h(t-1), x_t] + b_f)) \times f(W_o \cdot [H_h(t-1), x_t] + b_o).
\end{aligned}
\tag{9}
$$

After obtaining the indirect sensitivity and the direct sensitivity, the overall sensitivity is obtained by the following formula

$$S_h(t) = S_h^1(t) + S_h^2(t). \tag{10}$$

## Self-organizing LSTM algorithm

The self-organizing LSTM algorithm proposed in this research is based on the sensitivity analysis calculations. Figure 5 displays the flow of the algorithm.

First, randomly initialize the hidden layer's network parameters and neuron count. Next, the neural network's training phase begins. The network constantly engages in iterative training as long as neither the number of iterations nor the output value of the loss function has reached the predetermined threshold value of $a = 2$. This network's loss function is described as follows:

$$E(t) = \sqrt{\frac{1}{2t}\sum_{p=1}^{t}(y_d(p) - y(p))^2} \tag{11}$$

where $y_d(p)$ and $y(p)$ represent the expected output and actual output of the network at time p. In each iterative training, four values must be calculated first: output value $E(t)$

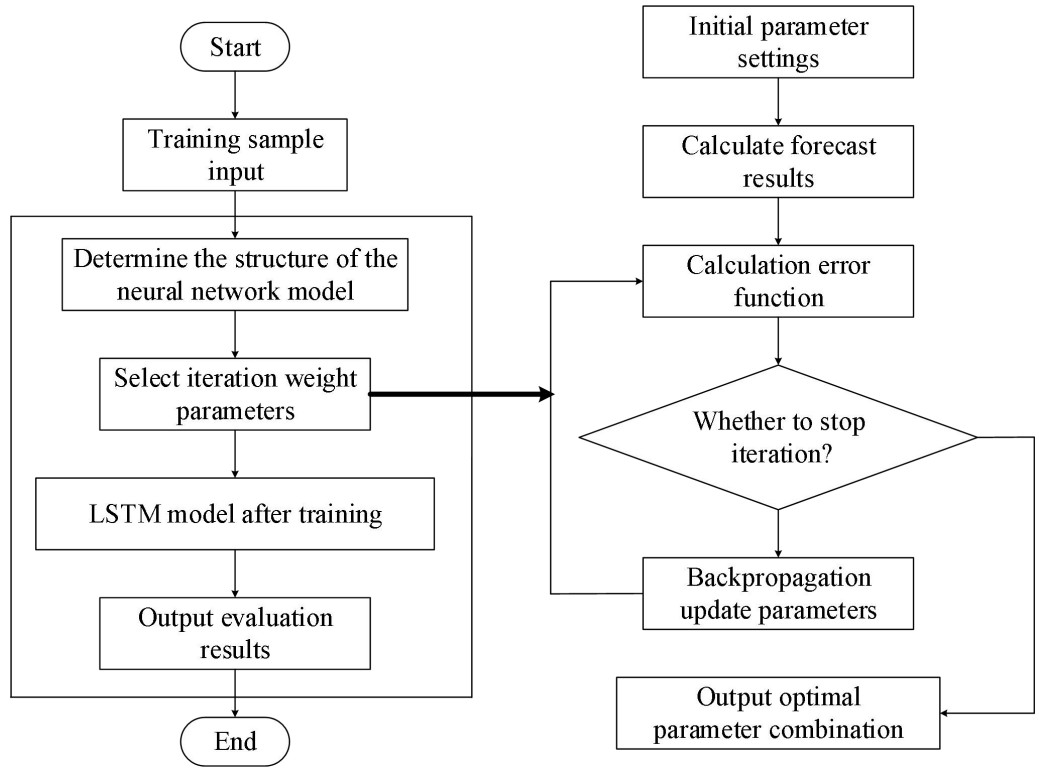

**Figure 5  Process based on LSTM risk assessment.**

of loss function (see (11)); indirect sensitivity $S_h^1$ of hidden layer neurons (see (5)); direct sensitivity $S_h^2$ of hidden layer neurons (see (8)); total sensitivity $S_h$ of hidden layer neurons (see (10)).

When $E(t) > \xi(t)$, it indicates that the performance of the network has not achieved the desired effect, and the N+1 LSTM unit needs to be added. Then, the weight initialization of the LSTM unit is as follows:

$$
\begin{aligned}
W_{N+1_x}^1(t) &= W_{n_x}^1(t) \\
W_{N+1_i}^1(t) &= W_{n_i}^1(t) \\
W_{N+1_f}^1(t) &= W_{n_f}^1(t) \\
W_{N+1_o}^1(t) &= W_{n_o}^1(t) \\
W_{N+1_s}^1(t) &= W_{n_s}^1(t) \\
W_{N+1}^2(t) &= W_n^2(t)
\end{aligned}
\tag{12}
$$

where $m$ represents the input weight of the new LSTM unit; $W_{N+1_i}^1$ represents the input gate control signal of the new LSTM unit; $W_{N+1_f}^1$ represents the forgetting gate control signal of the new LSTM unit; $W_{N+1_o}^1$ represents the output gate control signal of the new LSTM unit; $W_{N+1_s}^1$ represents the weight of the output self feedback loop of the new LSTM

unit; $W_{N+1}^2$ represents the connection weight of the unit to the output layer; $n$ represents the neuron with the highest total sensitivity among the existing hidden layer units.

In addition, when $S_n < \xi$, the m-th neuron needs to be deleted. The weights of this neuron are all 0, that is

$$W_{n_x}^1(t) = W_{n_i}^1(t) = W_{n_f}^1(t) = W_{n_o}^1(t) = W_{n_s}^1(t) = W_n^2(t) = 0. \tag{13}$$

At this point, the operation of adding and deleting the entire neuron is completed.

Figures 5 and 6 illustrate the fundamental steps involved in putting the LSTM-based risk assessment approach into practice.

The main steps of the algorithm proposed in this article are as follows:

**Step 1:** Data collection and preprocessing: Firstly, collect data related to rural microcredit, including borrower's personal information, historical loan records, repayment status, *etc.* Then, preprocess the data, including Data cleansing, missing value processing, Outlier detection, *etc.*, to ensure the quality and integrity of the data.

**Step 2:** Feature engineering: After pretreatment, feature engineering is required to extract the characteristics that have an impact on rural microfinance risk.

**Step 3:** Model training and optimization: Divide the dataset into training and testing sets, train the model using the training set, and evaluate the model using the testing set. In the training process, it may be necessary to adjust the super parameters of the model, such as Learning rate, batch size, *etc.*, to optimize the model performance.

**Step 4:** Risk assessment and prediction: After model training, the model can be used to conduct risk assessment and credit default prediction for new borrowers. By leveraging the predictive power of the model, it is possible to better control the default risk of rural microcredit and optimize lending decisions.

The self-organizing LSTM algorithm automatically adjusts the number of neurons by introducing self-organizing algorithms, thereby improving the performance of the model. Specifically, self-organizing algorithms dynamically increase or decrease the number of neurons during model training based on the characteristics and complexity of input data to optimize model performance.

The following are the steps for self-organizing LSTM to automatically adjust the number of neurons to improve model performance:

**Step 1:** Initial number of neurons setting: In the model construction phase, an initial number of neurons needs to be set first. This quantity can be a fixed value selected based on experience, or determined through some heuristic method or adaptive algorithm.

**Step 2:** Retraining the model: After increasing the number of neurons, the model needs to be retrained to learn new network structures and parameters. During the retraining process, the model will adapt to more complex data characteristics based on the number of new neurons.

**Step 3:** Performance monitoring: After the model is retrained, the self-organizing algorithm will evaluate the performance indicators of the model again. If performance improves, continue to maintain the current number of neurons. If there is no significant improvement in performance, the self-organizing algorithm may continue to increase the number of neurons until the model achieves satisfactory performance.

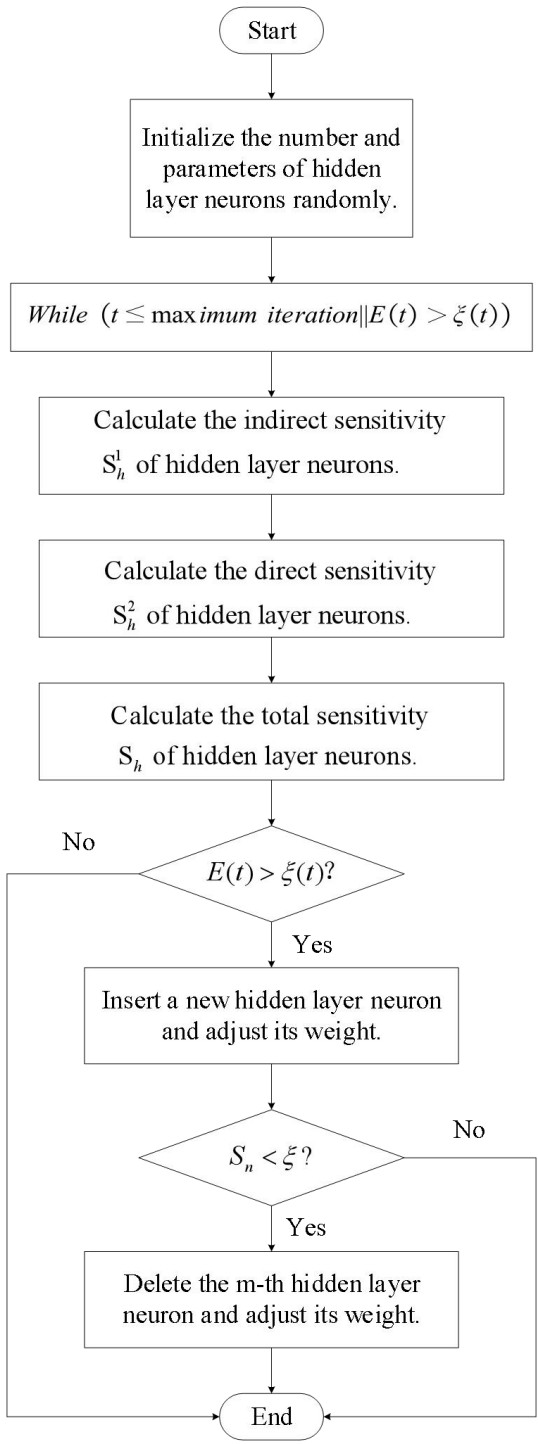

**Figure 6** **Self organizing LSTM algorithm flowchart.**

Through this self-organizing approach, self-organizing LSTM can dynamically optimize the number of neurons, enabling the model to better adapt to the needs of different scenarios

| Table 1 | Initial indicator variable. |
|---|---|
| Quantitative indicators | Borrower's age; contact number; The amount of the loan; Loan term; Family population; Supporting and supporting the population; Housing property; Production and operation property; Other properties (household cars, savings deposits, *etc.*); Cultivated land area; Annual agricultural income; Annual non agricultural income; Total household expenditure in the previous year; Amount of liabilities; Estimated total household income during the borrowing period; Estimated total household income during the borrowing period. |
| Qualitative indicators | Purpose of the loan; Education; Marital status; Bad credit record |

and datasets. This automatic adjustment mechanism helps to improve the performance of the model, making self-organizing LSTM perform better in tasks such as rural microcredit risk assessment.

# EXPERIMENTAL ANALYSIS

## Experimental preparation

The sample selected in this study is based on the data from a bank's microfinance database since 2008, and 166 samples are randomly selected. According to the five levels of classification of bank loans, risks are classified into five categories: normal, concerned, secondary, suspicious, and loss. If the principal or interest is overdue for 3–6 months during the supervision and use of the loan, the asset is recognized as a subprime loan, and the probability of loan loss is 30% to 50%. Therefore, in the selection of samples, loans with risks classified as sub prime and above are identified as default samples. We list the initial indicator variables in Table 1.

In this article, the samples are divided into default samples and non default samples according to whether they are in default, with 91 default samples and 75 non default samples. Training and testing phases make up the LSTM. The performance of the model is correlated with the amount of training samples. In the training stage, there are 112 samples, including 60 default samples and 52 non default samples; in the testing stage, there are 54 samples, including 31 default samples and 23 non default samples.

We are unable to get the dataset used in earlier investigations since the dataset for these studies contains consumer privacy information. Therefore, to gather and collate the dataset in the customer database, we employed the dimension definition method of the dataset indicated in earlier studies. Firstly, the technology proposed in this article was cross validated ten times based on the dataset compiled using the Khashman dimension definition method. The ten cross validation accuracy results are shown in Fig. 7.

Figure 7 demonstrates that the test group had a high classification accuracy rate, with an average classification accuracy rate of 92.97%.

As can be seen from Table 2, the addition and deletion of neurons are very frequent, and the sensitivity based self-organization algorithm is triggered multiple times during the training process, thus verifying the effectiveness of the self-organization algorithm.

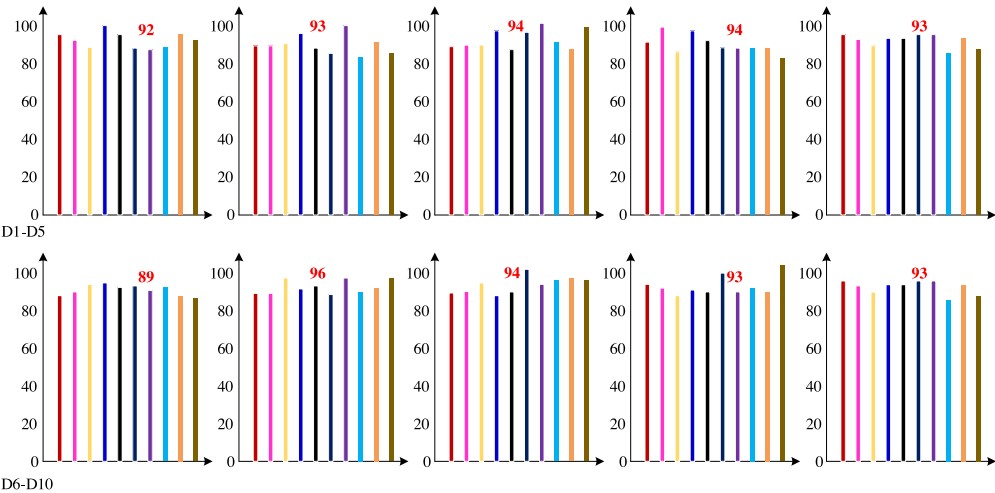

**Figure 7** **Ten cross validation accuracy results (%).**

**Table 2** **Quantitative analysis of different methods.**

|  | BP neural network | RNN | LSTM | Proposed method |
|---|---|---|---|---|
| Accuracy | 71.5% | 78.0% | 83.2% | 92.6% |
| Precision | 83,2% | 88.7% | 92.4% | 98.8% |
| Recall | 81.1% | 86.4% | 90.3% | 96.2% |
| F1 score | 78.2% | 79.1% | 82.3% | 89.2% |
| AUC value | 0.34 | 0.45 | 0.72 | 0.88 |

## Experimental comparison

We used BP neural network, RNN, LSTM, and the method suggested in this article for comparative experiments to demonstrate the superiority of the algorithm presented in this article.

First, the convergence rates of various techniques are contrasted, with the findings displayed in Fig. 8. There have been 675 iterations of the BP neural network algorithm, 411 iterations of the RNN algorithm, and 186 iterations of the conventional LSTM algorithm. The method outlined in this piece, however, achieves convergence after fewer than 100 iterations. As can be seen, the technique suggested in this article has a faster convergence rate than other methods.

Second, we test the algorithm's superiority using various data definition dimension techniques. The findings of ten times of cross-validation of these technologies using the Khashman's dimension defining method, Bekhet's dimension definition method, and

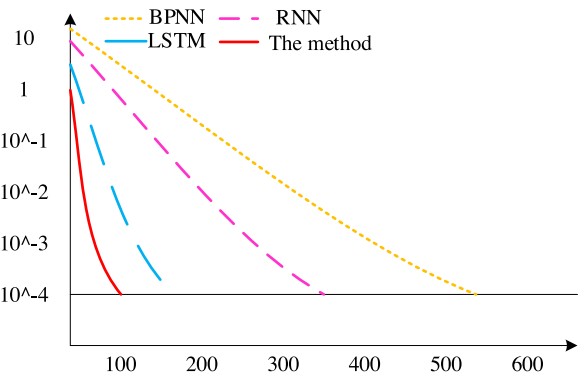

**Figure 8** Convergence speed of different algorithms.

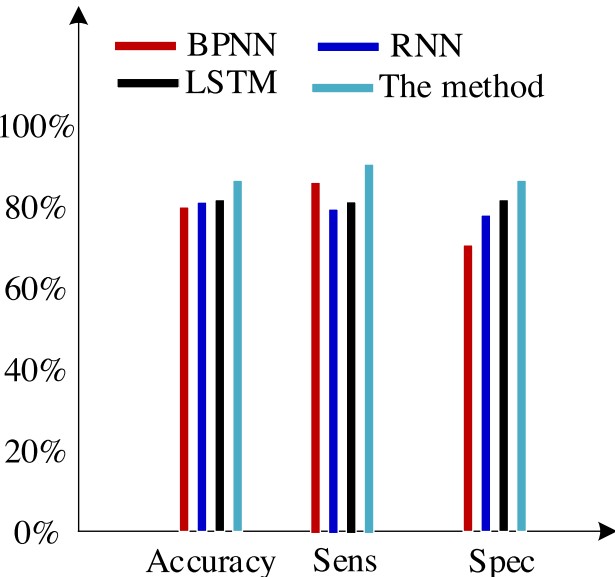

**Figure 9** Results of the trial based on the Khashman's dimensions definition technique.

the dimensions definition method in *Zhang et al. (2017)* are shown in Figs. 9, 10 and 11, respectively.

The method for dimension definition that we suggested makes up for the deficiencies of earlier datasets of consumer information by being comprehensive, objective, accurate, and suitable.

The three sets of experiments mentioned above were compared, and the comparison revealed the following analysis findings:

(1) There was no variation in the expression of customer information across the various data sets used in these experiments, even though they all contained collections of customer information.

(2) Khashman uses a wide range of dimensional definition methods, but they fall short in terms of efficiency and objectivity. For instance, determining whether an employee

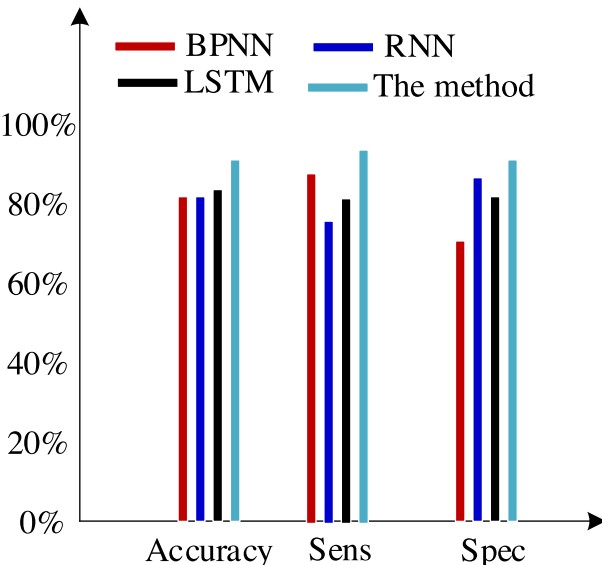

**Figure 10** Results of the trial based on Bekhet's dimensions definition technique.

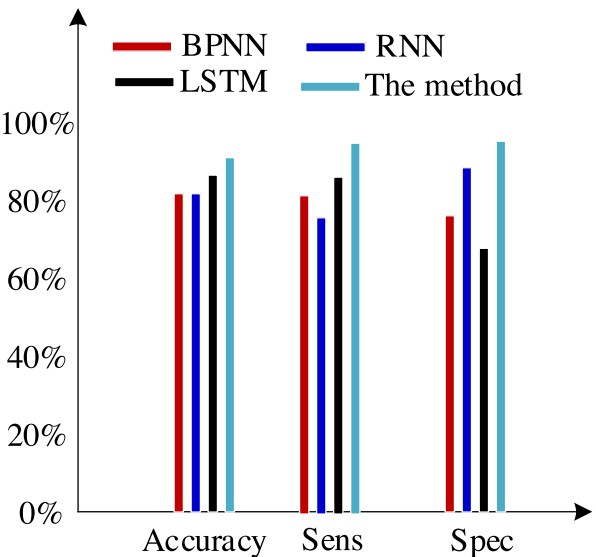

**Figure 11** Results of the experiment based on *Zhang et al. (2017)*'s dimensions definition technique.

is an overseas worker or not, or their phone number, can affect how their credit is evaluated. Therefore, the foundation for increasing classification accuracy is an objective and thorough dimension definition technique. Khashman's dimensional definition has been greatly improved by about 5% using the improved LSTM method suggested in this article. However, there is still space for development.

(3) *Bekhet & Eletter (2014)* suggested a more impartial approach to describing dimensions but did not support it. However, there is also a dearth of connection between

them because each dimension of the model is independent from the others. As a result, the classification accuracy of these algorithms has increased by almost 5% when compared to the tests based on the dataset compiled by the Khashman method, but there is still room for improvement.

(4) The literature *Zhang et al. (2017)* suggested a comprehensive, objective, and accurate method for defining dimensions, which compensated for the shortcomings of the above methods, to address the aforementioned issues. The classification accuracy of our suggested approach has increased by almost 5% as a result of the benefits of self-organizing LSTM models in learning the relationships between elements based on this dataset.

## Extended experiment

To compare the proposed self-organizing LSTM algorithm with previous neural network algorithms, we can select a dataset suitable for rural microcredit risk assessment and use a series of evaluation indicators to quantitatively evaluate the performance of each algorithm. The following is an extended experimental setup:

Evaluation indicators:

Select a suitable set of evaluation metrics to compare the performance of different algorithms. Common evaluation indicators can include:

Accuracy: The proportion of correctly predicted samples to the total sample size, which measures the overall accuracy of the model's prediction.

Precision: The proportion of true positive cases to predicted positive cases, measuring the accuracy of the model's prediction of positive cases.

Recall rate: The proportion of real cases to actual positive cases, measuring the model's ability to recognize positive cases.

F1 score: Taking into account both accuracy and recall metrics, a higher F1 score means that the model performs well in both accuracy and recall.

Receiver operating characteristic and AUC value: The Receiver operating characteristic represents the relationship between sensitivity and 1-specificity, and the AUC value represents the area under the Receiver operating characteristic, which is a comprehensive model evaluation index. Divide the selected dataset into training and testing sets. Train the model using self-organizing LSTM algorithm and other neural network algorithms, and tune each algorithm based on the validation set. Use a test set for model evaluation, and calculate the performance of each algorithm on evaluation indicators such as accuracy, precision, recall, F1 score, and AUC value. Compare the performance of various algorithms on different evaluation indicators, and analyze the advantages of self-organizing LSTM algorithm in rural microcredit risk assessment.

The experimental results show that the self-organizing LSTM algorithm outperforms other neural network algorithms in evaluation indicators such as accuracy, F1 score, and AUC value. This indicates that self-organizing LSTM can more accurately predict the credit status of borrowers and is more effective in risk assessment of rural microcredit. The self-organizing LSTM algorithm improves the performance and adaptability of the model by dynamically adjusting the number of neurons to better adapt to the needs of different scenarios and datasets.

## DISCUSSIONS

The method advanced in this article holds profound significance and substantial potential in practical applications.

(i) Exploring the risk assessment model of the enhanced LSTM algorithm can augment the precision and specificity of credit risk analysis, facilitating rural financial institutions in achieving superior control over default risk, minimizing credit losses, and propelling the steadfast growth of the rural microcredit market.

(ii) The integration of the risk assessment model with the refined LSTM algorithm can foster rural financial innovation, elevating the level of intelligence and personalization in financial services. This, in turn, shall ameliorate the financing landscape for farmers and expedite the modernization of the entire rural economy.

(iii) The improved model embraces a comprehensive consideration of distinctive risk factors specific to rural settings, encompassing aspects such as crop yields, weather fluctuations, and market dynamics. By doing so, it can accurately gauge the risks associated with rural microcredit, leading to the formulation of targeted risk management strategies.

The benefits for the overall development of the rural economy are manifold:

(i) Advancement of a stable rural economy, driving the upgrading and diversified evolution of rural industries, fostering increased employment opportunities, and mitigating the rural poverty rate.

(ii) Optimization of the allocation of financial resources, enhancing the efficiency and accessibility of rural financial services, and bolstering the resilience and competitiveness of the rural economy.

(iii) The incorporation of risk assessment models to mitigate default risk in rural microcredit and ensure financial stability contributes significantly to the sustainability and prosperity of the rural economy.

## CONCLUSION

The field of credit risk assessment is given a redesign and definition using a self-organizing LSTM model in this research. The method suggested in this research improves all evaluation indicators at various scales when compared to other methods, proving the value of using this LSTM model. According to the experimental findings, the technique suggested in this article has significant gains in accuracy and other areas in addition to having a faster convergence speed. future work should focus on further optimizing data collection and processing methods. There may be some difficulties in obtaining data in rural areas, and it is necessary to develop more intelligent and efficient data collection techniques while dealing with noise and missing values in rural data to improve the robustness and accuracy of the model.

### Funding
The authors received no funding for this work.

## Competing Interests

The authors declare there are no competing interests.

## Author Contributions

- Xia Gao conceived and designed the experiments, performed the computation work, prepared figures and/or tables, and approved the final draft.
- Xiaoqian Yang performed the experiments, prepared figures and/or tables, and approved the final draft.
- Yuchen Zhao analyzed the data, authored or reviewed drafts of the article, and approved the final draft.

## Data Availability

The data and code is available in the Supplemental Files.

The data are also available at Zenodo: Wesley Mendes Da Silva, Wilson Toshiro Nakamura, & Daniel C. Moraes. (2018). Data for "Credit card risk behavior on college campuses: evidence from Brazil" [Data set]. In Brazilian Administration Review (Vol. 9, Number 3, pp. 351–373). Zenodo. Available at https://doi.org/10.5281/zenodo.1303439.

## Supplemental Information

Supplemental information for this article can be found online at http://dx.doi.org/10.7717/peerj-cs.1588#supplemental-information.

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
