# Peer review of "Rural micro-credit model design and credit risk assessment via improved LSTM algorithm"

_PeerJ Computer Science, doi:10.7717/peerj-cs.1588_

## Round 0.1 · original submission · Major Revisions

Please see both reviewers' detailed comments. Reviewers highlighted that the paper should explain how the proposed model addresses inaccurate credit ratings and insufficient risk estimates. More details on the experimental setup, datasets, and evaluation metrics are needed, with specific quantitative results to support the claim of improved accuracy.

**Language Note:** The review process has identified that the English language must be improved. PeerJ can provide language editing services - please contact us at [email protected] for pricing (be sure to provide your manuscript number and title). Alternatively, you should make your own arrangements to improve the language quality and provide details in your response letter. – PeerJ Staff

Reviewer 1 ·

Basic reporting

In this study, combining qualitative and quantitative analysis methods, the LSTM method is used to establish a risk assessment model for small loans of farmers. According to the experimental results, the proposed algorithm is more accurate than the algorithm of recurrent neural network and conventional BP neural network.
Regarding the limitations and advantages of this study, the evaluation methods, results and validity of data interpretation need to be improved.

(1) Provide a brief background on the current state of farmers' micro loans and the existing challenges in the rural microfinance sector. This will provide context for the proposed research.

(2) Include a clear hypothesis or research question that the paper aims to answer regarding the credit risk of rural microfinance.

(3) Authors should add a paragraph into the introduction section. They should write "The main contributions of this paper are: (i) ….. (ii) ……. and (iii) ……" to highlight the key works. By this way, authors should provide a stronger motivation clearly and explain the originality of the paper.

(4) The second part, consistent in scope and value, focuses on the description of the models used and the analysis of the results.

(5) Explain how the proposed risk assessment model addresses the issue of inaccurate credit ratings and insufficient risk estimates. Discuss the specific features or mechanisms in the model that help improve accuracy and estimation.

(6) This paper is packed with lots of different equations and algorithms, but most of them are not clearly described.

(7) Expand on the experimental setup, including the datasets used and the evaluation metrics employed to compare the proposed algorithm with previous neural network algorithms. Present specific quantitative results to support the claim of improved accuracy.

(8) Clearly state the practical implications of the research and how the proposed risk assessment model can contribute to controlling default risk in rural microfinance. Discuss potential benefits for farmers, financial institutions, and the overall rural economy.

(9) The conclusion in its current form is confused in general. Concerning the Conclusion section, it would be better "Conclusions and Future Research", and it is strongly suggested to include future research of this manuscript. What will happen next? What are we supposed to expect from the future papers?

Experimental design

LSTM method is used to establish a risk assessment model for small loans of farmers. According to the experimental results, the proposed algorithm is more accurate than the algorithm of recurrent neural network and conventional BP neural network.

Validity of the findings

Regarding the limitations and advantages of this study, the evaluation methods, results and validity of data interpretation need to be improved.

Additional comments

Weaknesses of the article have been mentioned above. For each of the above, incorporate a detailed explanation in the article and your response.

Reviewer 2 ·

Basic reporting

This study proposes a sensitivity-based self-organizing LSTM algorithm to determine the ideal number of neurons in the hidden layer. The method described in this study demonstrates the usefulness of using this representation of customer information feature dimensions and the LSTM model, which enhances all evaluation metrics at different scales compared to other methods.
This is a good research paper for the present perspective. You all had a great effort overall. I appreciate it. Please follow the following comments to improve the quality of this article.

(1) The abstract should be constructed in a concise manner that presents readers with an instructive map to the paper. Abstract should be composed in a logical and accurate reflection of the organizational structure of the paper.
(2) Clarify the specific objectives and scope of the research at the beginning of the abstract. Clearly state the main problem addressed and the research contributions.

Experimental design

(3) Elaborate on the methodology used to establish the risk assessment model using the improved short-term memory (LSTM) algorithm. Provide an overview of the key steps involved and highlight any novel or innovative aspects.
(4) Provide a brief overview of the self-organizing LSTM algorithm proposed in the paper. Clearly explain how it addresses the problem caused by the number of traditional LSTM neurons and how it automatically adjusts the number of neurons.

Validity of the findings

(5) The linguistic quality needs improvement. It is essential to make sure that the manuscript reads smoothly- this definitely helps the reader fully appreciate your research findings. Consult a professional.
(6) Discuss the limitations of the proposed approach, including any potential drawbacks or challenges that may arise in its practical implementation. This will help provide a balanced perspective on the research findings.
(7) Consider providing recommendations for future research directions, such as exploring additional features or alternative algorithms to enhance further the accuracy and efficiency of risk assessment in rural microfinance.

---

## Round 0.2 · accepted · Accept

Both reviewers have confirmed that the authors have addressed all of thier comments.

Reviewer 1 ·

Basic reporting

The authors have successfully integrated all the comments. I am in agreement to proceed with accepting the article in its current state.

Experimental design

The methodology has been revised as requested.

Validity of the findings

The results section has also been revised as requested.

Reviewer 2 ·

Basic reporting

The paper has been revised in a good manner, the comments are considered and fulfilled

Experimental design

The required changes are made nicely.

Validity of the findings

The paper entirely looks good and can be accepted